# Dysregulated Antibody, Natural Killer Cell and Immune Mediator Profiles in Autoimmune Thyroid Diseases

**DOI:** 10.3390/cells9030665

**Published:** 2020-03-09

**Authors:** Tiphaine C. Martin, Kristina M. Ilieva, Alessia Visconti, Michelle Beaumont, Steven J. Kiddle, Richard J. B. Dobson, Massimo Mangino, Ee Mun Lim, Marija Pezer, Claire J. Steves, Jordana T. Bell, Scott G. Wilson, Gordan Lauc, Mario Roederer, John P. Walsh, Tim D. Spector, Sophia N. Karagiannis

**Affiliations:** 1Department of Twin Research and Genetic Epidemiology, King’s College, London SE1 7EH, UK; alessia.visconti@kcl.ac.uk (A.V.); chelle_mb@hotmail.com (M.B.); massimo.mangino@kcl.ac.uk (M.M.); claire.j.steves@kcl.ac.uk (C.J.S.); jordana.bell@kcl.ac.uk (J.T.B.); scott.wilson@uwa.edu.au (S.G.W.); tim.spector@kcl.ac.uk (T.D.S.); 2School of Biomedical Sciences, University of Western Australia, Crawley, WA 6009, Australia; 3Department of Oncological Sciences, Icahn School of Medicine at Mount Sinai, New York, NY 10029, USA; 4Tisch Cancer Institute, Icahn School of Medicine at Mount Sinai, New York, NY 10029, USA; 5St John’s Institute of Dermatology, School of Basic & Medical Biosciences, King’s College London, Guy’s Hospital, London SE1 9RT, UK; kristina.ilieva@kcl.ac.uk (K.M.I.); sophia.karagiannis@kcl.ac.uk (S.N.K.); 6Breast Cancer Now Research Unit, School of Cancer & Pharmaceutical Sciences, King’s College London, Guy’s Cancer Centre, London SE1 9RT, UK; 7Department of Biostatistics and Health Informatics, Institute of Psychiatry, Psychology and Neuroscience, King’s College, London SE5 8AF, UK; steven.kiddle@mrc-bsu.cam.ac.uk (S.J.K.); richard.j.dobson@kcl.ac.uk (R.J.B.D.); 8MRC Biostatistics Unit, University of Cambridge, Cambridge CB2 0SR, UK; 9Health Data Research UK (HDR UK), London Institute of Health Informatics, University College London, London NW1 2DA, UK; 10NIHR Biomedical Research Centre at Guy’s and St. Thomas’s NHS Foundation Trust, London SE1 9RT, UK; 11Department of Endocrinology and Diabetes, Sir Charles Gairdner Hospital, Nedlands, WA 6009, Australia; eemun.lim@health.wa.gov.au (E.M.L.); john.walsh@health.wa.gov.au (J.P.W.); 12Medical School, The University of Western Australia, Crawley, WA 6009, Australia; 13PathWest Laboratory Medicine, QEII Medical Centre, Nedlands, WA 6009, Australia; 14Genos, Glycoscience Research Laboratory, 10000 Zagreb, Croatia; mpezer@genos.hr (M.P.); glauc@genos.hr (G.L.); 15Faculty of Pharmacy and Biochemistry, University of Zagreb, 10000 Zagreb, Croatia; 16ImmunoTechnology Section, Vaccine Research Center, NIAID, NIH, Bethesda, MD 20892, USA; roederer@nih.gov

**Keywords:** multi-omic, autoimmune thyroid diseases (AITD), genetic variants, apoptosis, antibody-dependent cell-mediated cytotoxicity (ADCC), anti-thyroid peroxidase antibody (TPOAb)

## Abstract

The pathogenesis of autoimmune thyroid diseases (AITD) is poorly understood and the association between different immune features and the germline variants involved in AITD are yet unclear. We previously observed systemic depletion of IgG core fucosylation and antennary α1,2 fucosylation in peripheral blood mononuclear cells in AITD, correlated with anti-thyroid peroxidase antibody (TPOAb) levels. Fucose depletion is known to potentiate strong antibody-mediated NK cell activation and enhanced target antigen-expressing cell killing. In autoimmunity, this may translate to autoantibody-mediated immune cell recruitment and attack of self-antigen expressing normal tissues. Hence, we investigated the crosstalk between immune cell traits, secreted proteins, genetic variants and the glycosylation patterns of serum IgG, in a multi-omic and cross-sectional study of 622 individuals from the TwinsUK cohort, 172 of whom were diagnosed with AITD. We observed associations between two genetic variants (rs505922 and rs687621), AITD status, the secretion of Desmoglein-2 protein, and the profile of two IgG N-glycan traits in AITD, but further studies need to be performed to better understand their crosstalk in AITD. On the other side, enhanced afucosylated IgG was positively associated with activatory CD335^-^ CD314^+^ CD158b^+^ NK cell subsets. Increased levels of the apoptosis and inflammation markers Caspase-2 and Interleukin-1α positively associated with AITD. Two genetic variants associated with AITD, rs1521 and rs3094228, were also associated with altered expression of the thyrocyte-expressed ligands known to recognize the NK cell immunoreceptors CD314 and CD158b. Our analyses reveal a combination of heightened Fc-active IgG antibodies, effector cells, cytokines and apoptotic signals in AITD, and AITD genetic variants associated with altered expression of thyrocyte-expressed ligands to NK cell immunoreceptors. Together, TPOAb responses, dysregulated immune features, germline variants associated with immunoactivity profiles, are consistent with a positive autoreactive antibody-dependent NK cell-mediated immune response likely drawn to the thyroid gland in AITD.

## 1. Introduction

Autoimmune thyroid diseases (AITD) are a class of chronic, organ-specific disorders of the thyroid gland with a high genetic heritability (55–75%) [1,2,3,4] affecting approximately 5% of the population and with a gender disparity (i.e., women: 5–15%; men: 1–5%) [5,6,7]. Pathologically, AITD are characterized by autoantibodies against three main thyroid proteins (thyroid peroxidase (TPO), thyroglobulin (Tg), and the thyroid-stimulating hormone (TSH) receptor (TSH-R)), infiltration of the thyroid gland by immune cells (e.g., lymphocytes, NK cells, monocytes, and macrophages), the formation of germinal centers in the thyroid gland [8] and dysregulated TSH levels [9,10]. However, some studies have failed to observe a significant difference in peripheral blood immune cell composition between AITD patients and healthy individuals [11], while others report significant differences in particular cell types or in immune marker expression [12]. Immune cells, thyroid autoantibodies, and secreted proteins including cytokines may play critical roles in AITD development [13] and in immune responses, including in antibody-dependent cell-mediated cytotoxicity (ADCC) pathways [14,15]. However, the underlying autoimmune signatures associated with AITD remain unclear.

ADCC is triggered via antigen/antibody/Fc receptor complex formation, bringing the effector cell (macrophages, NK cells) and the target cell (expressing the antigen) in close contact. The formation and function of antigen/antibody complexes are modulated by various factors including post-translational modifications of glycans decorating antibodies [16,17]. One example is lack of fucose on the N-linked core glycan of IgG. Afucosylated antibodies have a higher affinity (~100-fold) for the immunoglobulin Fc receptor FcγRIIIa (CD16a), expressed on NK cells, macrophages and γδ T cells, and are shown to confer enhanced ADCC potential in vitro and anti-tumor activity in vivo [18,19,20,21]. This could result in antibodies with more potent Fc-mediated effector functions able to more effectively recruit and activate immune effector cells such as NK cells to kill target antigen-expressing cells [22,23]. IgG core fucose, observed in approximately 95% of IgG in healthy individuals, is considered a “safe switch” that can attenuate potentially harmful antibody-dependent damage against self-antigen-expressing normal tissues [18,19,20,21,24]. However, it is possible that these processes may be altered in autoimmune diseases.

We previously studied the glycosylation profiles of total immunoglobulin G (IgG) and of peripheral blood mononuclear cells (PBMC) in patients with AITD [4], as well as the glycosylation of IgG-depleted serum proteins in Hashimoto’s thyroiditis (HT) patients [25]. In peripheral blood, we identified both depleted core fucosylation of IgG antibodies and decreased antennary α1,2 fucosylation of PBMC to be associated with autoantibodies to thyroid peroxidase (TPOAb) and AITD status [4]. We also identified a network of genes, including *FUT8* and *IKZF1* that regulate fucosylation, to be implicated in the development of AITD [4,26]. Based on these findings, we speculated that IgG core fucose deficiencies together with elevated levels of autoantibodies may participate in autoimmune responses in AITD by enhancing effector cell activation and heightened immune and inflammatory signals.

Therefore, here we investigated immune features that may signify dysregulated, and likely heightened immune effector cells, antibodies, and immune mediators in AITD. In this in silico study in the blood of 622 subjects from the TwinsUK cohort, of whom 172 have AITD features, we aimed to investigate: (1) the association of different components of antigen/antibody/Fc receptor complexes with AITD; (2) the associations between these different immune components in a cohort of samples from volunteers regardless of disease status, and (3) potential genetic drivers on these components (study design summarized in Figure 1). Specifically, we examined the association of total serum IgG glycosylation, immune traits, such as immune cell subpopulation frequencies (CSFs; i.e., relative frequencies of circulating immune cell subsets), immune cell surface protein expression levels (SPELs; i.e., the measurement of the cell-surface expression of critical proteins) and secreted proteins, in the peripheral blood of patients with AITD compared with those of healthy volunteers (sample sizes of each study performed are summarized in Appendix A).

## 2. Materials and Methods

### 2.1. Study Sample 

The study was conducted using immune cell traits, glycosylation, proteomics, genotyping, and phenotypes in samples from research volunteers from the UK Adult Twin Registry (TwinsUK cohort, London, UK). The TwinsUK cohort is comprised of approximately 14,000 monozygotic and dizygotic same-sex adult twins from the UK, unselected for any particular disease or trait (Appendix A**)**. The cohort is of Northern European/UK ancestry and has been shown to be representative of singleton populations and the UK population in general [36,37]. Ethical approval was granted by the National Research Ethics Service London-Westminster, the St Thomas’ Hospital Research Ethics Committee (EC04/015 and 07/H0802/84). Informed consent was obtained from all study participants.

### 2.2. Data Statement

Multi-omic data were derived from samples in the TwinsUK cohort. Individual-level TwinsUK data, including phenotypes and genotypes, are not permitted to be shared or deposited due to the original consent given at the time of data collection. Access data can be applied for through the TwinsUK data access committee (http://twinsuk.ac.uk/resources-for-researchers/access-our-data/).

### 2.3. Definition of AITD and Detection of TSH and TPOAb

The study was performed using a clinical AITD definition and TPOAb as a threshold trait; it was not possible for AITD (Hashimoto’s disease and Graves’ disease) clinical diagnosis to be confirmed by a clinician. However, approximately 90% of individuals with Hashimoto’s disease, about 75% with Graves’ disease, <20% with other thyroid diseases, and <10% of normal individuals are known to have TPOAb-positivity [38,39,40]. Therefore, individuals were considered to have AITD if they either showed significantly higher than normal TPOAb serum titers (set at 3-fold higher than the threshold set by the manufacturer [18 IU/mL for the Abbott assay (ABBOTT Diagnostics Division, Wiesbaden, Germany) and 100 IU/mL for the Roche assay (Roche Diagnostics, Indianapolis, IN, USA)]) or had TSH serum levels >10 mIU/L. We considered individuals as controls if they had normal levels of TSH and a negative TPOAb titer, with no previous clinical diagnosis of thyroid disease and who were not treated with thyroid medications or steroids. Individuals with a history of thyroid cancer or thyroid surgery were excluded. Among the 622 individuals studied, 172 (27.65%) were identified with AITD, 236 (37.94%) considered normal controls, and 214 (34.41%) have TPOAb or TSH serum levels outside the normal range, but do not reach the 3-fold cutoff for inclusion in the AITD cohort. Evaluations of sera to measure TPOAb and TSH levels are described in Appendix B.

### 2.4. Detection of IgG Glycosylation Profiling for Discovery 

Plasma specimens for analysis of IgG glycosylation was collected between 1997 and 2013 in 2279 individuals from the TwinsUK cohort. IgG glycosylation profiling was performed on total plasma IgGs glycome (combined Fc and Fab glycans and all IgG subclasses) in Genos Glycoscience Research Laboratory (Zagreb, Croatia) using UPLC analysis of 2AB-labelled glycans (Waters Corporation, Milford, MA, USA). Protocol, data pre-processing and normalization in the TwinsUK cohort were previously described [4] (Appendix B). 

### 2.5. Detection of Immune Cell Traits 

Plasma samples for assessment of 78,000 immune traits were collected between 2010 and 2012 in 669 female participants from the TwinsUK cohort using high-resolution deep immunophenotyping flow cytometry analysis (BD BioSciences, San Jose, CA, USA) as previously described [28]. 78,000 different cell surface marker combinations captured by 7 distinct 14-color immunophenotyping panels were detected and described immune cell subset frequencies (CSF) and immune cell-surface protein expression levels (SPELs). After quality control to remove immune cell traits that appeared as poor reproducibility or out of range, 23,485 immune cell traits from 497 individuals of the TwinsUK cohort were analyzed. For this analysis, only 374 twins had immune cell traits data and TPOAb level detected by Roche immunoassay and 245 individuals in a case-control study by combining Roche and Abbott assays (204 controls and 41 AITD). Immune traits were quantile normalized residuals of a linear mixed effect model where age was included as fixed effects, and the batches were considered as random effects.

### 2.6. Detection of Protein Profiling in Plasma

With an aptamer-based multiplex protein assay (SOMAscan v2, SomaLogic Inc, Boulder, CO, USA) [41,42], 1129 proteins were measured (2013) on plasma samples collected between 2004 and 2011 from 211 female twins of the TwinsUK cohort (Appendix B). 

### 2.7. Statistical Analyses

All statistical analyses were run using R version 3.2.3. Linear mixed effect models were conducted using the R lme of package lme4 [43], and linear models were done in using R function lm of package stat. Custom R scripts developed for this study are available at this URL: https://github.com/TiphaineCMartin/multiomic_AITD.git. For determination of effective number of independent tests for different *-omic* data, association studies between *-omics* features and thyroid phenotypes and heritability analysis for proteins (Appendix B).

### 2.8. Genome-Wide Association Analysis on IgG N-Glycan Traits

To define genetic variants (i.e., single nucleotide polymorphisms (SNP), short insertions and deletions (indels)) associated with glycosylation profiles regardless of specific phenotypes in the TwinsUK cohort, we ran analyses with the GenABEL software package [44] designed for genome-wide association study (GWAS) analysis of family-based data by incorporating pairwise kinship matrix calculated using genotyping data in the polygenic model to correct relatedness and hidden population stratification. Data were recently published with other datasets [26,45]. We selected genetic variants for each IgG N-glycan traits with a *p*-value under the GWAS threshold (*p*-value < 5 × 10^−8^) and added the list of previously-defined genetic variants [29,45] (Appendix B). 

### 2.9. Determination of Shared Genetic Variants and Genes between IgG N-Glycan Traits, Immune Cell Traits, Protein Abundance, and Thyroid Functions and Diseases

To examine whether IgG N-glycan, immune cell traits, proteins, thyroid functions and diseases shared genetic variants or genes, we compared the genetic variants from GWASs on TwinsUK data (NHGRI GWAS catalog and other projects). As genetic variants detected by GWASs could be lead genetic variants but not necessarily causal genetic variants [46], we extended the list of genetic variants to other variants in linkage disequilibrium (LD) with an *r*^2^ threshold of 0.8 from 1000 G Phase 1 European population. Using HaploReg V4.1 [47] and GTeX data [32,33], we extracted tissue-specific expression quantitative traits (eQTLs) associated with these genetic variants.

### 2.10. Visualization

Heatmaps were created in using R package ComplexHeatmap. Correlation plots were created with R package corrplot. Boxplots and scatter plot were created in using R package ggplot2.

## 3. Results

### 3.1. Depletion of IgG Core Fucose is Positively Associated with Increased CD158b^+^ CD314^+^ CD335^−^ NK Cell Subset Counts 

IgG N-glycosylation is considered indispensable for the effector functions of IgG and inflammation control [48,49,50,51,52] and plays an essential role in the recognition and binding to Fc receptors of immune cells [51]. Using high-resolution deep immunophenotyping flow cytometry analysis in 669 twins from the TwinsUK cohort and IgG N-Glycan traits in 2297 twins from the same cohort [4,27,28], we identified 383 samples with measurements of 23,485 immune cell and 17 AITD-IgG N-glycan traits (IGP2, IGP7, IGP8, IGP15, IGP21, IGP33, IGP36, IGP42, IGP45, IGP46, IGP48, IGP56, IGP58, IGP59, IGP60, IGP62 and IGP63) and searched for any associations between them (Appendix A). 

In our cohort, we identified 1357 independent immune cell traits among 23,485 potential tested immune cell traits, where the partial correlation between immune cell traits is highlighted in Appendix A, 20 independent IgG N-glycan traits among 75 potential tested IgG N-glycan traits, and 6 independent AITD-IgG N-glycan traits among 17 potential tested AITD-IgG N-glycan traits [53]. Association studies of total IgG N-glycan traits with immune cell traits showed that 6 of the 17 identified significant IgG N-glycan traits (IGP2, IGP42, IGP46, IGP58, IGP59, IGP60) previously associated with TPOAb level and AITD status in the TwinsUK cohort, were also associated with 51 immune cell traits, which are all NK cells (CD16^+^ CD56) featuring different combinations of 6 immunoreceptors (CD2, CD158a, CD158b, CD314, CD335, R7) (Appendix A, Figure 2). Three IgG N-glycan traits without core fucose (IGP2, IGP42, IGP46) were negatively associated with the level of the activating subpopulation of CD16^+^ CD56^+^ CD158b^−^ CD335^+^ NK cells and positively associated with the level of the CD16^+^ CD56^+^ CD335^−^ effector NK cell subpopulation and with the activating subpopulation CD16^+^ CD56^+^ CD158b^+^ CD314^+^ CD335^−^ NK cells [54,55,56,57,58,59]. In contrast, three other significant IgG N-glycan traits with core fucose (IGP58, IGP59, and IGP60) had the opposite effect associations with the same subpopulations of NK cells (Figure 2a). In agreement with our previous report, there are therefore negative correlations between the set of IgG N-glycan traits without core fucose (IGP2, IGP42, IGP46) and the set of IgG N-glycan traits describing IgG core fucose (IGP58, IGP59, and IGP60) [4] (highlighted in Figure 2b). Moreover, we observed strong correlations between these 51 immune cell traits (Figure 2c). The presence of correlation patterns between the 17 AITD-IgG N-glycan traits (Figure 2b) as well as between the 51 immune cell traits (Figure 2c) is consistent with our observation of correlations between the 6 AITD-IgG N-glycan traits and the 51 immune cell traits (Figure 2a). When we extended our analysis to the 58 remaining IgG N-glycan traits also identified in our samples, but not associated with AITD, we observed no significant association between them and the 23,485 immune cell traits. Moreover, for 23,485 peripheral blood immune cell traits (Appendix A), no significant association with AITD or TPOAb level could be identified (Appendix A).

We conclude that a subpopulation of NK cells (CD16 + CD56) and specifically the activating subpopulation CD16^+^ CD56^+^ CD158b^+^ CD314^+^ CD335^−^ NK cells is associated with fucose-depleted IgG in individuals with AITD.

### 3.2. The AITD-Associated Genetic Variants, rs1521 and rs3094228, Alter Thyroid Cell Expression of Ligands for CD314 and CD158b Immunoreceptors

The NK cell receptors, CD335 (NKp46), CD314 (NKG2D) and the killer cell immunoglobulin-like receptors (KIRs) including CD158b, are normally associated with activated NK cell states, T cell co-stimulation, and mediating tumor cell lysis [55,57]. To determine whether genetic factors could contribute to AITD, related immune features, or their pathways, we inspected genetic variants associated with AITD, TPOAb levels, and immune cell traits from previous GWAS [27,28,30]. We then compared these with recent large-scale studies on tissue-specific expression quantitative traits (eQTLs) [35], mainly from the GTEx project [32,33,34] in blood and thyroid tissue. 

No genetic variants previously associated with AITD or other thyroid phenotypes appeared to be associated with the expression of CD335 or its known ligands in blood and thyroid cells. However, we observed that two genetic variants, rs1521 and rs3094228, associated with Graves’ disease and TPOAb-positivity, respectively, fall in the gene regulatory regions of *MIC-A* and *MIC-B* genes, two ligands of CD314 (NKG2D), and alter their gene expressions in thyroid cells [32,33,34,60,61,62] (Figure 3, Appendix A, Appendix A). These two AITD-genetic variants, rs1521 and rs3094228, also alter the expression of the *HLA-C* gene, ligand of CD158b, in thyroid cells. The Graves’ disease (GD) risk allele of rs1521 variant is primary associated with a reduced expression of *HLA-C* gene, ligand of CD158b, in thyroid cells. Furthermore, the TPOAb-positivity risk allele of rs3094228 variant is primary associated with an increased expression of *MIC-B* gene, ligand of CD314 (NKG2D), in thyroid cells. As about 75% of patients with Graves’ disease have TPOAb-positivity and rs3094228 that has been associated with TPOAb-positivity and Graves’ disease [61,63], it is possible that the association of rs1521 with Graves’ disease could be also driven by TPOAb-positivity and, so, associated with its phenotypes. Downregulation of *HLA-C* gene expression and upregulation of *MIC-A* and *MIC-B* gene expression in thyrocytes could activate NK cell functions and the cytokine production against thyrocytes when NK cells and thyrocytes are in contact. 

Furthermore, three genetic variants, rs2596460, rs2596457 and rs2523691, previously associated with higher levels of the subpopulation of NK cells featuring CD16^+^ CD56/CD2^−^ CD314^+^ CD335^−^ CD337^−^ CD158a^+^ CD158b^+^ [28], are in the same haplotype as the rs3094228 genetic variant, but with a low linkage disequilibrium (LD, *r*^2^ < 0.8) (Appendix A). Potentially, one of the genetic variants in this locus are the causal genetic variant of higher abundance of CD158b^+^ CD314^+^ CD335^−^ NK cells. All of the three genetic variants could also alter the expression of the *HLA-C* gene, ligand of CD158b, and *MIC-A*, ligand of CD314 (NKG2D), in immune cells [32,33,34,64] (Figure 3, Appendix A). 

Overall, two genetic variants, rs1521 and rs3094228, associated with, respectively, Graves’ disease and TPOAb-positivity, appear to alter thyrocyte expression of ligands of two immunoreceptors of NK cells, CD314 and CD158b; both of which have the capacity to enhance cytotoxicity of NK cells after binding with target cells. Additionally, three genetic variants in the same haplotype than rs3094228 could increase the abundance of the immune active CD158b^+^ CD314^+^ CD335^−^ NK cell subpopulation.

### 3.3. AITD is Associated with Increased Serum Caspase-2 and IL-1α

We next evaluated whether the abundance of 1113 free soluble proteins, which are partially correlated between each other (Figure 4a), in peripheral blood may be associated with AITD status (27 AITD patients versus 130 healthy controls) and TPOAb levels (155 individuals of whom 25 have AITD) in the TwinsUK cohort (Appendix A) using aptamer-based multiplex protein assay (SOMAscan) [68]. Firstly, we observed significant moderate correlations of the TSH levels measured by two clinical-certificated assays (Abbott and Roche) with the TSH levels measured by the SOMAscan assay (Figure 4b).

This indicated that the SOMAscan assay could reproduce with good accuracy the estimation of TSH levels and probably also for other proteins. Levels of three proteins were positively associated with AITD status (Bonferroni multiple testing correction, *p*-value < 1.9 × 10^−4^): TSH (*p*-value = 8.67 × 10^−5^; Beta = 0.67; SE = 0.16), Caspase-2 (CASP-2; *p*-value = 2.72 × 10^−7^; Beta = 1.10; SE = 0.20) and Interleukin-1α (IL-1α; *p*-value = 7.46 × 10^−5^; Beta = 0.41; SE = 0.09). We also observed higher mean levels of TSH in patients with AITD (mean_SOMAscan_ = 1443.9, sd_SOMAscan_ = 1238.5; mean_clinical_ = 7.1 IU/mL, sd_clinical_ = 10.47) or TPOAb-positivity (mean_SOMAscan_ = 1389.9, sd_SOMAscan_ = 1018.3; mean_clinical_ = 5.7 IU/mL, sd_clinical_ = 7.55) compared with controls (euthyroidism with TPOAb-negative) (mean_SOMAscan_ = 851.1, sd_SOMAscan_ = 368.6; mean_clinical_ = 1.64 IU/mL, sd_clinical_ = 0.79). Although Caspase-2 and IL-1α levels were associated with AITD status, Caspase-2 and IL-1α levels were not associated with TPOAb or TSH levels as continuous variables (*p*-value > 1.9 × 10^−4^). However, when participants were divided into 4 categories according to TSH and TPOAb levels (Figure 4c), reflecting different clinical categories (hyperthyroidism, euthyroidism/TPOAb-negative, hypothyroidism and euthyroidism/TPOAb-positive), Caspase-2 showed significantly higher mean and variance in two groups: hypothyroidism and euthyroidism/TPOAb-positive (Figure 4d). The hypothyroidism and euthyroidism/TPOAb-positivity in this cohort potentially indicate underlying Hashimoto’s thyroiditis (HT). This is because HT is the most common cause of hypothyroidism, spontaneous hypothyroidism (i.e., no previous history of thyroid ablation) is almost always caused by HT, and euthyroid individuals with TPOAb-positivity almost always have HT when studied by cytology and histopathology [69,70,71]. On the other hand, the variance of IL-1α was significantly larger in groups with euthyroidism/TPOAb-positive and hyperthyroidism (Figure 4e), but there was no significant difference for their mean values. Hence, individuals from 4 categories have the same levels of IL-1α, but there are more inter-individual variabilities in euthyroidism/TPOAb-positive and hyperthyroidism than euthyroidism/TPOAb-negative and hypothyroidism. 

In summary, we confirmed the association of the plasma TSH levels with AITD status, and we found two novel associations of plasma Caspase-2 and IL-1α with AITD status, but their secretion (mean and variance) seems to also depend on other factors associated with thyroid diseases such as the levels of TSH and TPOAb.

### 3.4. Afucosylated IgG is Associated with Serum Levels of Several Circulating Proteins

When we studied the correlation between the level of secreted TSH, Caspase-2 and IL-1α proteins and IgG N-glycan trait levels in 164 individuals of whom 27 have AITD, we found no significant associations (*p*-value > 8.3 × 10^−4^, Bonferroni test considering 3 independent proteins and 20 independent IgG N-glycan traits) (Appendix A, Figure 4f). However, several AITD-IgG N-glycan traits appeared to be associated with 7 other circulating proteins (AMHR2, BCMA, β2-microglobulin, ERBB1, Desmoglein-2, TRAILR4, and FCGR3B) (*p*-value < 3.67 × 10^−5^, Bonferroni test in considering only 227 independent proteins and 6 independent IgG N-glycan traits) (Appendix A, Figure 4f). For example, three AITD-IgG N-glycan traits (IGP2, IGP42, and IGP46) were positively associated with circulating FCGR3B (FcγRIIIb or CD16b), an Fc receptor expressed by polymorphonuclear neutrophils (PMN), whereas three AITD-IgG N-glycan traits (IGP58, IGP59, and IGP60) were negatively associated with the antibody Fc receptor FCGR3B. Also, IGP56 and IGP48 were negatively associated with β2-microglobulin, involved in the presentation of intracellular antigens through the MHC class I complex; and IGP48 was positively associated with ERBB1, the epidermal growth factor receptor (EGFR), a checkpoint molecule associated with cellular proliferation and differentiation.

Overall, 12 AITD-IgG N-glycan traits (IGP2, IGP8, IGP42, IGP46, IGP48, IGP56, IGP57, IGP58, IGP59, IGP60, IGP62, and IGP63) were associated with serum levels of 7 circulating proteins (AMHR2, BCMA, β2-microglobulin, ERBB1, Desmoglein-2, TRAILR4, and FCGR3B) in the TwinsUK cohort.

### 3.5. Free-Soluble Plasma Desmoglein-2 Protein is Associated with AITD Genetic Variants and Two AITD-IgG N-Glycan Traits

We evaluated several GWAS on secreted proteins (protein quantification locus traits, pQTL) [31], to determine whether the secretion of proteins associated with AITD or with AITD-IgG N-glycan traits are driven by AITD genetic variants. We found no genetic variants associated with any of 17 AITD-IgG N-glycan structures that are also pQTL. However, four genetic variants associated with thyroid phenotypes published in the GWAS catalog (rs3761959, rs7528684, rs505922, and rs3184504) were also associated in *cis* and *trans* with nine circulating protein abundances (BGAT, CHSTB, DC-SIGN, Desmoglein-2, DYR, FCRL3, GP1BA, MBL, and VCAM-1) (Appendix A). None of these proteins were associated directly with AITD or TPOAb levels in our study. However, we found that Desmoglein-2 was associated with two AITD-IgG N-glycan traits, IGP8, and IGP63 [4] (Appendix A). Desmoglein-2 is highly expressed in epithelial cells including thyrocytes and cardiomyocytes and plays a role in the cell-cell junctions between epithelial, myocardial, and certain other cell types and is thought to be a regulator of apoptosis [74]. 

Therefore, four genetic variants associated with thyroid phenotypes are also associated with nine secreted protein abundances, including the apoptosis regulator Desmoglein-2 in blood. Desmoglein-2 was also associated with two AITD-IgG N-glycan traits.

## 4. Discussion

The dysregulation of the immune system may affect several biological structures and processes in AITD, such as antigen/antibody/Fc receptor complex formation, possibly driven by genetic and environmental factors [75]. Little is known about the key players and the genetic variants identified in previous GWASs of patients with AITD. Targeting of self-antigen expressing tissues by immune cells may depend on the formation of antigen/antibody/Fc receptor complexes featuring substantial affinity or avidity properties. In the peripheral blood of individuals with AITD, we previously detected depletion in IgG core fucose that is known to enhance such interactions and may influence immune effector cell engagement of target cells by antibodies. We proposed that this signature is associated with TPOAb levels and with immune effector cell activation in patients with AITD [4]. Here, we reveal immune and genetic features pointing to activated NK cell subsets, thyroid cell-derived ligands for immunoreceptors on NK cells, alongside secreted mediators of apoptosis and immune activation, all signals of heightened antibody and innate effector cell responses in AITD. 

We applied an in silico multi-omic approach on peripheral blood specimens from individuals from the TwinsUK cohort to investigate any association between immune features and genetic variants in AITD. In AITD patient samples, we observed increased levels of three circulating proteins (TSH, Caspase-2, and Interleukin-1α) and a decreased level of IgG core fucosylation associated with an activated subpopulation of NK cells defined primarily by the expression of CD335, CD134, and CD158b receptors. Our data confirms the previously reported association of plasma TSH level with AITD status and also reveals previously unknown potential biomarkers for AITD, which are highly associated with immunological activation functions, such as ADCC, apoptosis and pro-inflammatory pathways. Furthermore, several genetic variants previously associated with AITD appear to alter thyrocyte gene expression of several ligands of NK immunoreceptors and abundance of plasma circulating proteins. This suggest that the genetic background may also play potential roles in NK cell activation likely focused on thyroid cells in individuals with AITD. 

To our knowledge, no other cohorts have large datasets that are available to interrogate and feature the same diversity of -*omics* data with AITD phenotype or TPOAb levels. In our studies, we note an imbalance in the sample sizes between control individual groups and AITD groups. This is because the dataset comes from unselected twins and reflects the general European population [37], where approximately 5% of the population, but 5–15% for women, present individuals with AITD [5,6,7]. To overcome such imbalances in our sample sizes and low samples sizes with large *-omics* data, we applied machine learning and non-parametric methods with correction for multiple testing. Another limitation in our present study is the absence of AITD clinical diagnosis confirmed by clinicians for all individuals. We consequently applied more stringent criteria to define patients with AITD versus control individuals, by using TSH and TPOAb levels (see Section 2.3 of our Materials and Methods). We also performed analysis on TPOAb levels, as this is considered the main clinical quantitative biomarker of AITD status [38,39,40]. Replication and meta-analysis studies on larger *-omic* datasets incorporating clinical features will help to confirm our present findings.

Two secreted proteins (Caspase-2 and IL-1α), which play roles in apoptosis and the inflammatory response, were positively associated with AITD. TPOAb have been proposed to target thyroid cells by engaging effector cells via their Fc receptors [4,14,15,76,77], and the apoptosis protein Caspase-2 may represent a marker potentially signifying antibody-mediated destruction of thyroid cells [78]. In concordance, IL-1α, produced by activated immune, epithelial and endothelial cells in response to cell injury and apoptosis, is considered an apoptosis index of the target cell [79] and proportional to the degree of lymphoid infiltration in thyroid disorders [80]. IL-1α seems to reduce the thyroid epithelial barrier, even in the absence of any other signs of cytotoxicity [81]. In concordance, in our study we found higher levels of secreted IL-1α in AITD blood compared with levels in healthy individuals, and its variance was greater in euthyroidism/TPOAb-positive blood and in hyperthyroidism. This may signify dysregulation in cellular structures in the thyroid gland. Overall, Caspase-2 and IL-1α may reflect the degree of thyroid cell death or apoptosis and of lymphoid infiltration towards the thyroid gland. 

A subpopulation of NK cells expressing combinations of immunoreceptors (CD2, CD158a, CD158b, CD314, CD335, R7) was associated with the depletion of IgG core fucose in individuals with AITD. These included an activating NK receptor (CD314) and a differentiation receptor (CD335); whilst, fucose-depleted IgG was also positively associated with a subpopulation of NK cells with an inhibitory NK receptor (CD158b) [54,55,56,57,58,59,82]. The combination of potentially autoreactive antibodies with enhanced Fc domains and activated effector cells such as NK cells may signal increased inflammation and susceptibility to autoimmune disease [83]. Previous studies showed that afucosylated antibodies have a much higher affinity (100-fold) for FcγRIIIa (CD16a) and may thus have enhanced ADCC [84]. Moreover, ADCC via FcγRIIIa may require NK cells, but not monocytes or polymorphonuclear cells, and activity levels of the antigen/antibody/effector cell complexes have been correlated only with the NK cell numbers present in the PBMC [20]. Our associations between the levels of IgG core fucose and of a subpopulation of NK cells reinforce the notion that there is a complementarity between IgG core fucose levels and NK cells, that could influence effector cell potency, potentially against a range of antigens including self-antigens.

It has been previously estimated that AITD are highly heritable (55–75%) and that most of IgG N-glycan traits and the immune cell traits associated with AITD are moderately heritable (Appendix A) [4,27]. By estimating the proportion of genetic and environmental variance of 1129 proteins in our study using the Structural Equation Modeling and twin structures present in the TwinsUK cohort (Appendix A), we found a small proportion of proteins having additive genetic variances in their heritability, in concordance with previous findings on a smaller dataset [85]. As the best model of heritability in AITD is only with dominant genetic variance, the shared genetic variance between AITD and proteins as well as with IgG N-glycan traits and immune cell traits could not be estimated with accuracy. However, in our study, we identified several genetic variants previously associated with thyroid phenotypes to be also associated with the secretion of proteins and gene expression of ligands of two NK cell immunoreceptors. Specifically, genetic variants, rs1521 and rs3094228, associated with Graves’ disease and TPOAb-positivity, alter the expression of thyroid cell-expressed ligands, *MIC-A, MIC-B,* and *HLA-C*, known to recognize CD314 and CD158b immunoreceptors expressed on NK cells. Moreover, rs3094228 falls in the same European haplotype as three genetic variants associated with higher abundance of the activated CD158b^+^ CD314^+^ CD335^−^ NK cell subset. Thus, individuals having the AITD-risk allele of rs1521 variant have reduced expression of *HLA-C* gene and, at a lesser extent, expression of *MIC-A* in thyrocytes, whereas the carriers of AITD-risk allele of rs3094228 genetic variant associated with TPOAb-positivity showed increased expression of *MIC-B* gene in thyrocytes and potentially higher abundance of the highly active CD158b^+^ CD314^+^ CD335^−^ NK cells. Consequently, if the thyrocytes in carriers of AITD-risk alleles for CD158b and CD314 ligands crosstalk with the subpopulation of NK cells with CD158b and CD314 immunoreceptors with the help of the antibodies, they could trigger the production of cytokines and cytotoxicity against thyrocytes by these NK cells. 

## 5. Conclusions

Our findings thus highlight different immune features (glycan structures on antibodies, a subpopulation of immunoactive NK cells, the secretion of Caspase-2 and IL-1α) as potential signals of AITD status detectable in the bloodstream in addition to TSH and TPOAb levels. Moreover, if one speculates that active antibodies with low core-fucose might be thyroid autoantibodies (e.g., TPOAb) [86] and target cells are thyroid cells, it is conceivable (Figure 5) that that immune cell-antibody-target cell interactions may lead to cytotoxicity functions targeting thyroid tissues [76,86,87]. Together, these may form part of a dysregulated autoimmune response in AITD. Further replication studies and validation studies of real-time functional evaluations associated with these immune features and genetic analyses are needed to confirm this model. These features could also be tested in the context of thyroid cancer immunotherapy [77] in future studies.

## Figures and Tables

**Figure 1 cells-09-00665-f001:**
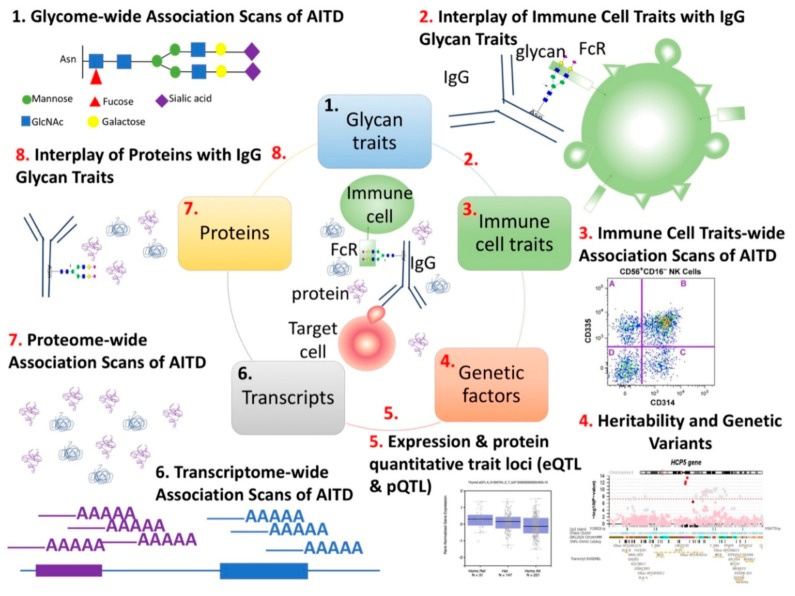
Multi-omics computational analyses were used to study the components of antigen/antibody/effector cell complex structure in AITD. (1) We previously performed glycome-wide association studies of AITD and TPOAb levels using 3146 individuals from three European cohorts, including the TwinsUK cohort. We identified 17 AITD-IgG N-glycan traits in the discovery TwinsUK cohort, and seven of these 17 have been then replicated in two other cohorts [4]. (2) In the present study, we studied the association of total IgG N-glycan traits with 23,485 immune cell traits in 383 individuals from the TwinsUK cohort (regardless of disease status). We showed that 6 out of the 17 AITD-IgG glycan traits were correlated with 51 immune cell traits featuring the CD335, CD134, and CD158b receptors. (3) None of these 51 immune cell traits appeared to be associated with AITD in 374 individuals (34 with AITD). (4) The heritability of AITD, TPOAb level and several -*omic* features (IgG N-glycan traits and immune cell traits) were performed in previous studies of the TwinsUK cohort [4,27,28,29]. Here we estimated the heritability of secreted proteins, but we could not determine shared additive genetic variance between different phenotypes studied (AITD status, TPOAb level, level of IgG N-glycan traits, of immune cell traits and of circulating proteins in the bloodstream). (5) We identified genetic variants that alter the expression of genes, proteins and cell-bound immune receptors (highlighted in this study) using the previous GWASs performed in the TwinsUK cohort or from GWAS catalog, eQTLs from GTEx project and pQTLs from INTERVAL project [27,28,30,31,32,33,34,35]. (6) We previously performed transcriptome-wide association studies of AITD, TPOAb level, and N-glycan structures in the whole blood of approximately 300 individuals and we found no significant associations [4]. (7) We observed 3 out of 1113 circulating proteins tested in plasma of almost 300 individuals shown to be associated with AITD status (TSH, Caspase-2, and Interleukin-1α). (8) Several secreted proteins were correlated with the level of plasma IgG glycan traits in 164 individuals, but none of them were also associated with AITD. The sample sizes of these different studies are described in Appendix A. GlcNAc = N-acetylglucosamine. The numbers in black depict analyses performed previously [4,27,28,29] while the numbers in red depict analyses presented for the first time in the present study.

**Figure 2 cells-09-00665-f002:**
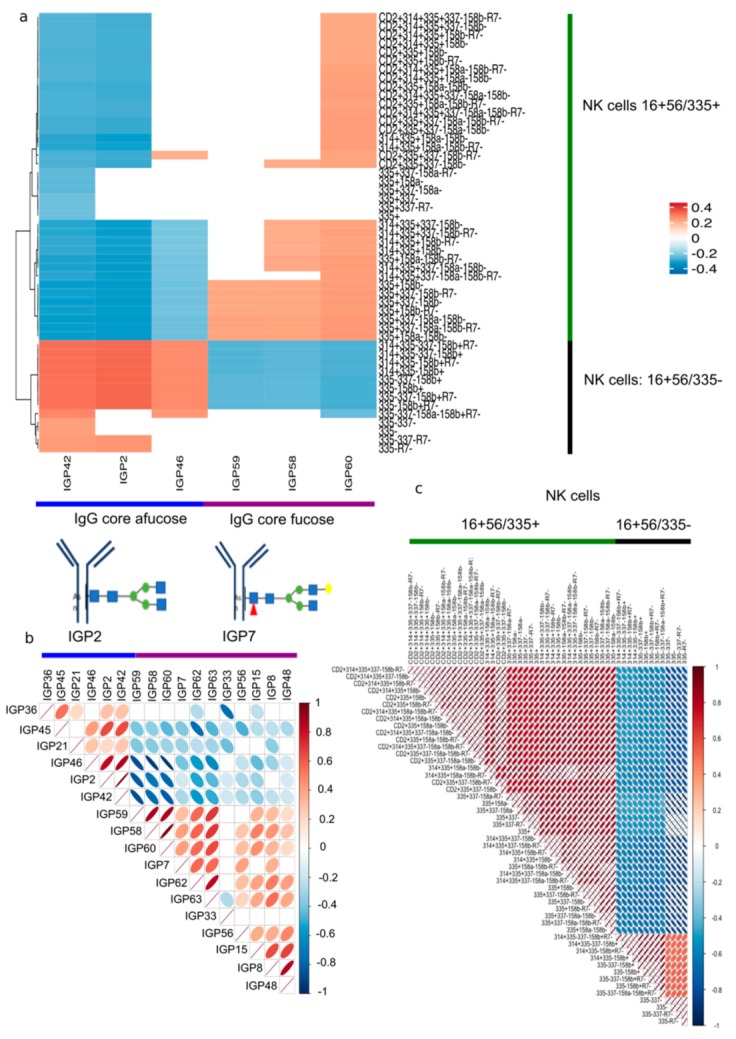
AITD-IgG N-glycan traits associated with a subpopulation of NK cells. (**a**) Heatmap of immune cell traits associated with AITD-IgG N-glycan traits. The 51 NK cell types were significantly associated with six out of 17 AITD-IgG N-glycan traits previously identified [4]. Below the heatmap, there are one representative of IgG core afucose (IGP2) and one representative of IgG core fucose (IGP7), that were both associated with AITD and TPOAb levels [4]. (**b**) Co-expressions between only 17 IgG N-glycan traits previously associated significantly with AITD status and TPOAb level [4]. (**c**) Correlations between the profile of 51 immune cell traits that were associated significantly with at least one of 17 AITD-IgG N-glycan traits. The order of immune cell traits is the same as that in Figure 2a.

**Figure 3 cells-09-00665-f003:**
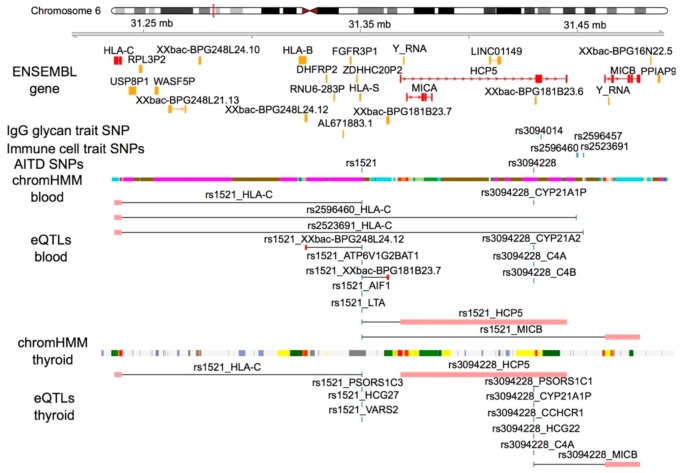
Association of immune cell traits with AITD status. Annotation tracks around *MIC-A, MIC-B* and *HLA-C* genes visualize significant GWAS hits for immune cell traits, the ligands of certain immunoreceptors (such as NK), and thyroid phenotypes previously identified in the TwinsUK cohort as well as chromatin states identified using chromHMM from whole blood from ENCODE [65] and thyroid cells from CEMT [66] and eQTLs from GTEx project [32,33]. The plot was produced using functions from R packages Gviz and coMET [67].

**Figure 4 cells-09-00665-f004:**
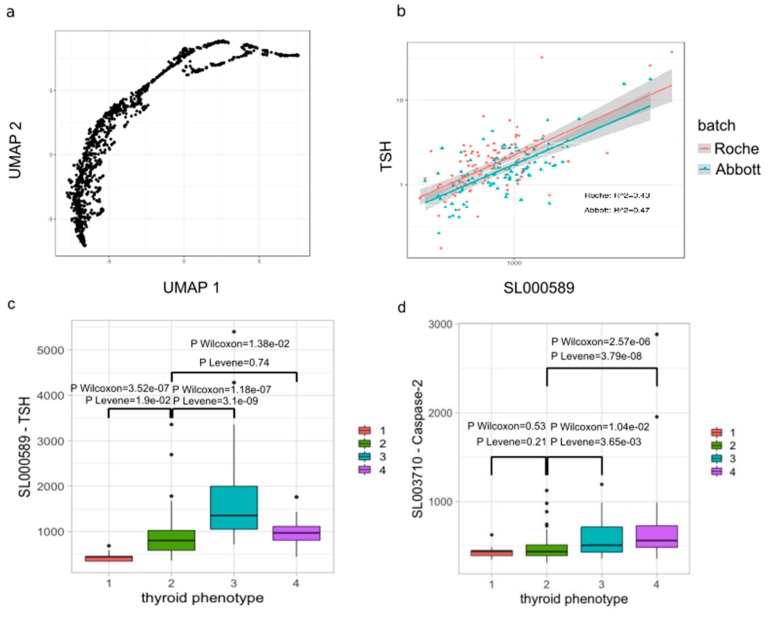
Association of circulating protein abundances with thyroid diseases and with AITD-IgG N-glycan structures. (**a**) 1,113 circulating proteins were arranged in two dimensions based on the similarity of their secretion profiles in the serum by the dimensionality reduction technique UMAP [72] using R package umapr [73]. (**b**) Correlation of log10-transformed TSH measurements between two clinical FDA approved clinical immunoassays (Roche and Abbott) and SOMAscan assay in 217 individuals (122 using Roche immunoassay and 95 using Abbott immunoassay). (**c**) Box plot of the level of circulating TSH measured by SOMAscan assay in the serum according to the group of thyroid status. (**d**) Box plot of the level of circulating Caspase-2 measured by SOMAscan assay in the serum according to the group of TSH. (**e**) Box plot of the level of circulating IL-1α measured by SOMAscan assay. An extreme outlier sample in the group 4 with an IL-1 α of 250,000 mg/mL was discarded for the analysis. (**f**) Heatmap of circulating protein abundances associated with AITD-IgG N-glycan structures. In Figure 2c–e, participants were assigned to 4 categories according to TSH level and TPOAb status: 1 = hyperthyroidism (TSH < = 0.1 mIU/L; 13 individuals), 2 = euthyroidism/TPOAb-negative (0.4 < TSH > 4 mIU/L & TPOAb < 6 IU/mL (Abbott) or TPOAb < 34 IU/mL (Roche); 196 healthy individuals), 3 = hypothyroidism (TSH > = 4 mIU/L; 21 individuals), and 4 = euthyroidism/TPOAb-positive (0.4 < TSH > 4 mIU/L & TPOAb > = 6 IU/mL (Abbott) or TPOAb > = 34 IU/mL (Roche); 28 individuals). Wilcoxon-Mann-Whitney’s test has been performed between groups to estimate whether there are mean differences whereas Levene’s test has been performed between groups to estimate whether there are variance differences.

**Figure 5 cells-09-00665-f005:**
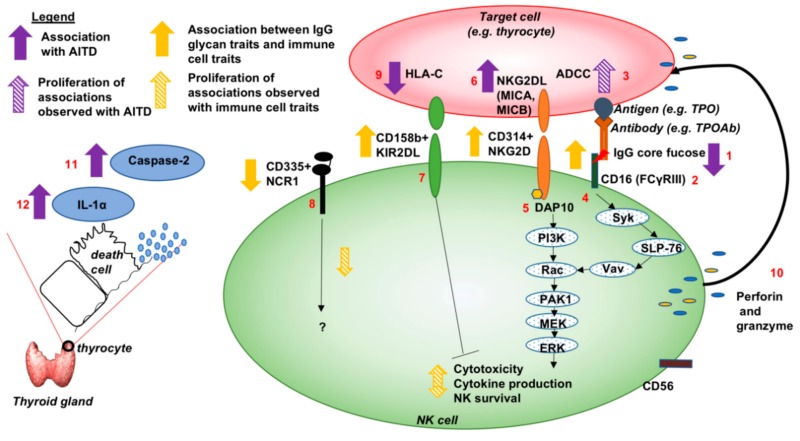
Model of different potential contributing players and their pathways activated in proposed antibody-dependent NK cell-mediated cytotoxicity in the thyroid gland of AITD patients. (1) The depletion of IgG core fucose was associated with TPOAb level and AITD status [4]. (2) The IgG N-glycan traits associated with AITD were also associated with a subpopulation of NK cells in our current study; for example, the depletion of IgG core fucose is associated positively with NK cells with the patterns of co-receptors CD335^−^ or CD335^−^ CD158b^+^ CD314^+^. (3) Previous studies showed that afucosylated antibodies had increased affinity for binding to CD16 (FcγRIIIa), cell receptors of NK cells, and to enhance ADCC [18,19,20,21] via (4) protein tyrosine kinase-dependent pathways, through crosstalk with (5) NKG2D receptor (CD314) [88,89]. (6) Two SNPs, rs3094228 and rs1521, were associated with GD and TPOAb-positivity [60,61,62] and fall in gene regulatory regions of the *MIC-A* and *MIC-B* genes and increase their expression in thyroid cells [32]. These two genes encode heavily glycosylated proteins that are ligands for the NKG2D type II receptor (CD314). (7) The KIR2DL (CD158b) receptor is known to regulate the cytotoxicity of NK cells by unknown pathways, whereas (8) the NCR1 (CD335) receptor can contribute to the increased potency of activated NK cells to mediate cell lysis by unknown pathway [54,55]. (9) The SNP, rs1521 associated with GD [60], is also shown to reduce the expression of HLA-C gene, producing the ligand of CD158b, in thyroid cells [32,33,58,59]. (10) All together (the binding of NK cells with target cells through antibodies and their ligands), these lead to the activation of NK cells, which release cytotoxic granules containing perforin and granzymes. This release mediates ADCC of target cells (3), which are thyrocytes in AITD. Also, (11) a positive association between the circulation abundance of Caspase-2 protein and AITD were found in this study that could be associated with the destruction of thyrocytes. (12) A positive correlation of circulating abundance of IL-1α with AITD was also found in the bloodstream that could be a marker of lymphocyte infiltration in the thyroid gland of individuals with AITD, and thus of inflammation [80,81].

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
