# Peer review of "Dysregulated Antibody, Natural Killer Cell and Immune Mediator Profiles in Autoimmune Thyroid Diseases"

_cells, 2020, doi:10.3390/cells9030665_

Round 1

Reviewer 1 Report

There are discrepancies and controversial results concerning immune cell populations composition in peripheral blood of AITD patients and healthy controls. The specific autoimmune signatures associated with AITD remain unclear, therefore the authors have performed an excellent study that significantly broadens the current state of knowledge. The manuscript is well written. The introduction is pertinent and the study design is summarized in a form of informative graph presenting subsequent stages of the analyses. The examined groups of patients varied depending on the type of particular studies and were derived from approximately 14 000 individuals from TwinsUK cohort, being still sufficiently numerous. Out off 622 individuals from the cohort, 172 were identified as AITD. Such a large group of patients provides the solid foundation for reliable, statistically significant results and conclusions. Statistical methods were based on complex mathematical models and algorithms. Moreover, authors used a wide range of state-of-the-art multi-omic in-silico techniques and modern cross-sectional studies, investigating specific immune features, protein and genetic variants in the AITD subjects included. The obtained results have been visualised in a form of well-designed heatmaps.

Please note that in Fig. 2A – the scale of the presented heatmap (on the right) is unintentionally cut off, possibly due to the size of the image.

The presented findings are original; they highlight potential biomarkers of possible AITD, detectable in the bloodstream, such as specific populations of immunoactive NK cells or the secretion of Caspase-2 and IL-1α.

Finally, authors proposed not only the brief summary of the obtained results, but also  conceivable and interesting model of molecular pathways and receptors involved in NK cell-mediated cytotoxity in the thyroid of AITD patients (see: Fig. 5).

I find the number of performed analyses impressive and I fully recommend the paper for publication in CELLS.

Author Response

March 3rd, 2020

We thank the Reviewers for their enthusiasm for our manuscript titled “Dysregulated antibody, natural killer cell and immune mediator profiles in autoimmune thyroid diseases” and for their comments and suggestions for improving the manuscript.

In response, in our revised manuscript, we have improved the description of our previous findings about IgG glycosylation in the Introduction, our findings related to genetic variants altering the gene expressions of ligands of NK cells in the Results section, and eventually their potential role in AITD in the Discussion. We acknowledge that further studies need to validate our proposed biomarkers for AITD, and the potential mechanisms proposed here. Our point-by-point responses to the Reviewers’ comments are summarized below.

Authors’ Response:

We thank Reviewer #1 very much for their positive comments. We rescaled the Figure 2A to avoid the partial cut off of the image

Reviewer 2 Report

It is an interesting work, very well performed. 

A lot of data is available regarding a possible role IgG core fucose deficiencies  in autoimmune responses in AITD probably by enhancing  effector cell activation profiles. 

Author Response

March 3rd, 2020

We thank the Reviewers for their enthusiasm for our manuscript titled “Dysregulated antibody, natural killer cell and immune mediator profiles in autoimmune thyroid diseases” and for their comments and suggestions for improving the manuscript.

In response, in our revised manuscript, we have improved the description of our previous findings about IgG glycosylation in the Introduction, our findings related to genetic variants altering the gene expressions of ligands of NK cells in the Results section, and eventually their potential role in AITD in the Discussion. We acknowledge that further studies need to validate our proposed biomarkers for AITD, and the potential mechanisms proposed here. Our point-by-point responses to the Reviewers’ comments are summarized below.

Authors’ Response:

We thank the Reviewer #2 very much for their positive review and comments.

Reviewer 3 Report

This article is scientifically relevant as the authors present new data on the immune mechanisms underlaying AIDT. The relationship between fucose-depleted IgG fucosylation and specific immunoreceptors signature in NK cells is very interesting, providing a new approach to the role of these cells in this type of disease. Likewise, the associations established with the secreted immune mediators and the genetic traits are promising , but I find them less well substantiated. In this sense, the study group of the AIDT is very small (n=13), moreover, some data necessary to establish the associations that are included in the conclusions are not provided (for example those related to thyrocytes, rs1521 and MIC A expression). A number of questions and concerns need to be revised prior publication.  

- The abstract is difficult to understand. Lines 46-47 describe the specific objective of the work and place the IgG core fucose deficiency in the core of the study. Directly following, lines 48 to 51, the objective is reformulated in a less precise way and this is quite confusing. The sentence in lines 57-58 (”…, alongside…”) is particularly confusing.

- Introduction section: more background is needed. More information should be included about previous results which are then continuously referred to throughout the paper, e.g. concerning IgG N-glycan traits.

- The objectives stated in the introduction need to be better explained and more specified. It is confusing that line 97 specifies the number of individuals in the study and that it is repeated in subobjective 2 but not in the others.

-Figure 1 has too much information. It is confusing to see that the numbers do not follow a correlative order. It is difficult to identify whether the studies indicated are previous results or those of the present work. This figure is introduced in the main text as a summary of the study design, but results are included in a confusing way. For example, in line 114, the statement headed "3)" refers to 374 individuals but it is not understood which study group they belong to.

-I have not found the supplementary tables (S1 to S9). Line 136 indicates Table s1, the heading of which appears in line 510, but the table does not appear.  The link indicated in line 487 to access the supplementary materials does not work.

- Reference 4 is very misleading to me. It appears constantly in the text and is indicated as a reference that endorses published results when, in fact, it is a bioRxiv preprints of the work presented, and not indexed by Web of Sci. For example, line 167, line 237, line 494.

-Line 224: Fig.S1a is indicated as if illustrating the results summarized in lines 224-226 and this figure does not show this information

-Lines 246-248: These lines express the conclusion of this section of results (3.1) and include a specific immunoreceptor signature that appears for the first time in the text. It should be highlighted in the figure 2 and indicated earlier in the text.

- Lines 269-270: Fig.S2 only shows information from tissue-specific expression studies of HLAC, MICB and HCP5 but not MIC-A. Therefore, no conclusions should be drawn regarding MIC-A. Furthermore, in relation to rs1521 and thyrocytes there is only information from HLAC , and no data are shown for either MIC B or MIC A. Therefore, no conclusion should be drawn regarding its expression related to this genetic variant in thyrocytes.

- Lines 296-323: In the Results section 3.3. When participants are divided into 4 categories, the number of individuals in each group is quite unequal. So in group 1 of hyperthyroids there are only 13 individuals. This number is too low to be able to draw really extrapolable conclusions. This number should be increased and the conclusions reformulated in results and discussion to be less categorical (lines 391-392). A discussion paragraph should be included with the limitations of the study.

-Line 307-310 : This refers to data that I have not found reflected in the paper, such as the mean levels of TSH in patients with AITD.

-Line 313: “caspase-2 showed significantly higher mean …”   I understand these are statistically significant differences, but I haven't found the pvalue

-Lines 313-314: “…and euthyroidism/TPOAb-positive (Fig. 4d), potentially  indicating underlying Hashimoto’s thyroiditis (HT).” I don't understand why this relationship with HT is established.

- Lines 342-358: This section explains the results of Figure 4f, but does not refer to the Desmoglein and TRAILR4 proteins that are included in the figure. They should be commented on or removed from the figure.

- Line 398: "…thyrocyte secretion of several ligands…" Only gene expression data on thyrocytes have been presented, one should not speak of protein secretion.

-Line 440-442: "genetic variants, rs1521 and rs3094228, ...., alter the expression of thyroid cell-expressed ligands, MIC-A, MIC-B, HLA-C,..." and line 445-446: "the variant rs1521 have reduced expression of HLA-C gene, but increased 445 expression of MIC-B in thyrocytes ..." The expression of MICA in thyrocytes is not analyzed at any time ( Fig. S2 ), and in relation to rs1521 only HLA C in thyrocytes and MIC B and HCP5 in peripheral blood are studied. Thus, this conclusion is not supported.

Minor points

Line 58: use abbreviation TPOAb

Line 62: Keywords.complete the name

Figures 1 and 2: image cut on the right edge

Figure 2a: IGP2, IGP7 should be incorporated next to the graphic schemes

Figure 2c: Increase the text size, cannot be read.

Line 85-86: “ in peripheral blood” (redundant)

Line 380: GWASs without the s

Figure 5: Format of the figure caption

Author Response

March 3rd, 2020

We thank the Reviewers for their enthusiasm for our manuscript titled “Dysregulated antibody, natural killer cell and immune mediator profiles in autoimmune thyroid diseases” and for their comments and suggestions for improving the manuscript.

In response, in our revised manuscript, we have improved the description of our previous findings about IgG glycosylation in the Introduction, our findings related to genetic variants altering the gene expressions of ligands of NK cells in the Results section, and eventually their potential role in AITD in the Discussion. We acknowledge that further studies need to validate our proposed biomarkers for AITD, and the potential mechanisms proposed here. Our point-by-point responses to the Reviewers’ comments are summarized in the attachment.
